

# *MaqFACS* (Macaque Facial Action Coding System) can be used to document facial movements in Barbary macaques (*Macaca sylvanus)*

Églantine Julle-Danière[1], Jérôme Micheletta[1], Jamie Whitehouse[1], Marine Joly[1], Carolin Gass[1], Anne M. Burrows[2,3] and Bridget M. Waller[1]

[1] Department of Psychology, Centre for Comparative and Evolutionary Psychology, University of Portsmouth, UK
[2] Department of Physical Therapy, Duquesne University, PA, USA
[3] Department of Anthropology, University of Pittsburgh, PA, USA

## ABSTRACT

Human and non-human primates exhibit facial movements or displays to communicate with one another. The evolution of form and function of those displays could be better understood through multispecies comparisons. Anatomically based coding systems (Facial Action Coding Systems: FACS) are developed to enable such comparisons because they are standardized and systematic and aid identification of homologous expressions underpinned by similar muscle contractions. To date, FACS has been developed for humans, and subsequently modified for chimpanzees, rhesus macaques, orangutans, hylobatids, dogs, and cats. Here, we wanted to test whether the MaqFACS system developed in rhesus macaques (*Macaca mulatta*) could be used to code facial movements in Barbary macaques (*M. sylvanus*), a species phylogenetically close to the rhesus macaques. The findings show that the facial movement capacity of Barbary macaques can be reliably coded using the MaqFACS. We found differences in use and form of some movements, most likely due to specializations in the communicative repertoire of each species, rather than morphological differences.

## INTRODUCTION

Human and non-human primates share a similar multi-modal communication system using auditory (e.g., vocalizations), olfactory (e.g., smells) and visual channels (e.g., gestures, facial expressions). Facial communication in primates is highly complex, involving subtle and dynamic facial movements (*Van Hooff, 1967*) and the evolution of this communication system may be linked to the increased complexity of their social system (*Freeberg, Dunbar & Ord, 2012*). A comparative approach is necessary to fully understand the evolution of this facial communication system and to identify any species-unique characteristics of the human face. However, analysis of those facial movements has not always been straightforward because of the lack of appropriate standardized and objective measurement tools (*Waller & Micheletta, 2013*).

Submitted **15 June 2015**
Accepted **27 August 2015**
Published **15 September 2015**

Corresponding author
Jérôme Micheletta,
jerome.micheletta@port.ac.uk

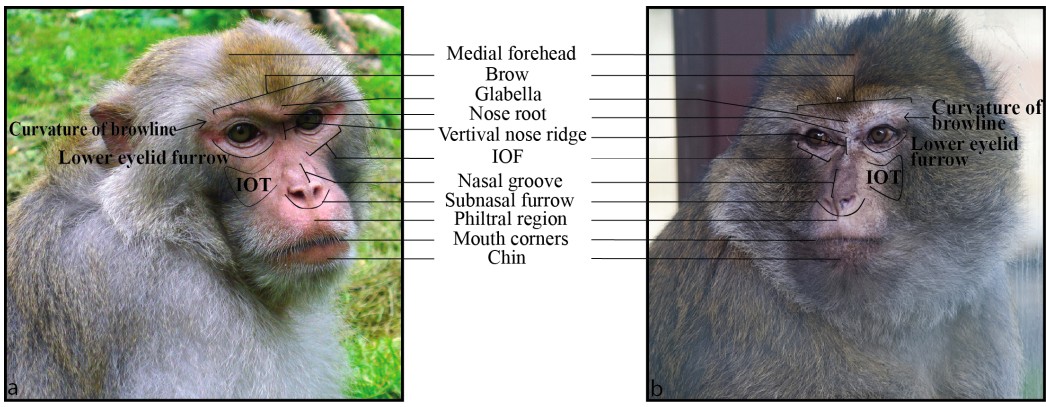

**Figure 1 An illustration of key facial landmarks for Barbary (B) and rhesus (A) macaques.** The Barbary macaque present a slightly more elongated face; the key facial landmark are the same in both species but the nasal groove and the vertical nasal ridge are longer in the Barbary compared to the rhesus macaques. Finally, the phitral region is more pronounced in the Barbary macaques. On the all, the Barbary macaques are bigger and present a more abundant fur, making it difficult to see the ears and their movements. Photos by Jérôme Micheletta (Rhesus macaques) and Jamie Whitehouse (Barbary macaques).

*Hjortsjo (1970)* and Ekman and colleagues (*1978*) were the first to document the facial movements in humans with reference to the underlying physiology. The development of the Facial Action Coding System (FACS, *Ekman & Friesen, 1978*; *Ekman, Friesen & Hager, 2002b*), an anatomically-based system describing facial movement in humans according to contraction of underlying facial muscle, allowed researchers to tackle previous methodological issues. For example, FACS is able to compare facial expressions objectively across individuals regardless of the inherent variability in the surface morphology of faces, e.g., bone structure, fatty deposits, skin texture, and individual muscle variations (*Waller et al., 2007*; *Waller, Cray & Burrows, 2008*). Since its creation, it has become the most widely used coding system in facial expression research, and requires training and certification to be used. FACS uses numbers to refer to 33 facial muscle contractions (Action Units, hereafter AUs) and 25 more general head/eye movements (Action Descriptors, hereafter ADs). Each AU and AD is presented in a manual with a name (e.g., AU9 for nose wrinkle), basic appearance changes with reference to basic morphological features/landmarks of the face (Fig. 1), and minimal criteria for identification. Because the system is based on muscles that vary little within species (*Waller, Cray & Burrows, 2008*), FACS can compare facial movements regardless of superficial individual differences in other aspects of facial anatomy, such as hair covering, facial coloration, bone structure, etc. This latter characteristic also makes FACS ideal for modification across species.

Development of the previous modified system followed a clear process. First, the presence, size, and structure of facial muscles were investigated using dissection (*Burrows, Waller & Parr, 2009*; *Burrows et al., 2006*). Second, the surface movements of individual muscles were demonstrated using intramuscular stimulation techniques (*Waller et al., 2008*; *Waller et al., 2006*). Third, the contraction of specific muscles was identified from

video footage of spontaneous behaviour, and the surface appearance changes described and compared in detail for documentation in the training manuals (e.g., http://www. orangfacs.com/). However, some FACS systems have been adapted from the Human FACS without following this 3-step procedure: the GibbonFACS (*Waller et al., 2012*), DogFACS (*Waller et al., 2013*), and EquiFACS (*Wathan et al., 2015*) were developed based on dissection and observation of spontaneous behaviours only. For ethical reasons, intramuscular stimulation is avoided unless there is the opportunity to use an existing planned procedure under anaesthesia for the procedure (see ChimpFACS and MaqFACS for details; *Vick et al., 2007*; *Waller et al., 2008*). Intramuscular stimulation does provide additional information but is not essential.

To date, FACS has been modified for use with chimpanzees (*Pan troglodytes*: ChimpFACS (*Vick et al., 2007*)), rhesus macaques (*Macaca mulatta*: MaqFACS (*Parr et al., 2010*)), gibbons (Hylobatids, GibbonFACS (*Waller et al., 2012*)), orangutans (*Pongo* spp: OrangFACS (*Caeiro et al., 2013*)), dogs (*Canis familiaris*: DogFACS *Waller et al., 2013*), cats (*Felis catus*: CatFACS (CC Caeiro, AM Burrows & BM Waller, unpublished data)), and horses (*Equus caballlus*: EquiFACS (*Wathan et al., 2015*)). Each FACS system is based on the human FACS, so that individual movements can be directly compared between species. The development process itself, therefore, is highly informative because the process can reveal how similar the target species is (in terms of the capacity for facial movement) to the previous species under study.

However, inter-genus studies are not the only ones that could give us an insight into the evolution of facial expressions. Intra-genus studies are also important to help understand species-specific expressions, where species differences could be linked to different social systems. GibbonFACS is a first example of intra-genus studies resulting in a modified system allowing the study of facial movement in Hylobatids (gibbons and siamangs, *Waller et al., 2012*). The genus *Macaca* is another excellent group for intra-genus comparisons, offering even more intra-genus variation in social styles (*Thierry, 1990*; *Thierry, 2007*). At one extreme are the more socially tolerant species such as Tonkean macaques (*M. tonkeana*). These species are characterized by relatively relaxed dominance styles with tolerance toward subordinates, low levels of aggression, high levels of reconciliation and affiliation and a low kin bias (*Balasubramaniam et al., 2012*; *Sueur et al., 2011*; *Thierry & Aureli, 2006*; *Thierry et al., 2008*). At the other extreme are the 'more despotic' species such as Japanese macaques (*M. fuscata*) and rhesus macaques (*M. mulatta*). These species are characterized by mainly unidirectional conflicts and reduced conciliatory tendencies, with a rigid dominance hierarchy (*Thierry & Aureli, 2006*). Any affiliations tend to be kin-based (*Aureli, Das & Veenema, 1997*). These differences in social style can and have been linked to differences in communication (*Dobson, 2012*; *Preuschoft, 1995*).

In both tolerant and despotic species, individuals must communicate with one another and this is done mainly via vocalizations and visual displays such as facial movements. Both the vocal and facial display repertoires of some macaque species are relatively well understood, especially in rhesus macaques (*M. mulatta*). Facial displays and vocalizations in this species convey information related to the rank of the sender, individual identity,

reproductive status and emotional state/intent of the signaller (*Andrew, 1963*; *Gerald, Waitt & Little, 2009*; *Preuschoft, 2000*; *Redican, 1975*; *Van Hooff, 1962*). The Barbary macaque (*M. sylvanus*) is unique within its genus in two obvious respects: (1) it is the only macaque species that is distributed outside of Asia, and (2) it is the most ancient taxon of the genus (*Purvis, 1995*). Barbary macaques appear intermediate in dominance style: they are relatively tolerant but with an existent hierarchy (*Thierry & Aureli, 2006*).

Thus, it would be extremely useful to have a FACS system available for use with Barbary macaques in order to make inter specific comparisons. Barbary macaques strongly resemble the rhesus macaques, with the most marked difference that the Barbary macaques have more abundant fur, covering both their faces and bodies (Fig. 1). As the phylogenetic conservation of facial muscles is high from Old World monkeys to New World monkeys to apes (*Burrows, 2008*), it is highly likely that the facial musculature of Barbary macaques strongly resembles that of rhesus macaques. Based on this strong resemblance, we wanted to test whether the Macaque Facial Action Coding System (MaqFACS), developed for rhesus macaques, could be used to code facial movements in Barbary macaques without significant modification from the original system.

## METHODS

### Subjects

We collected footage from a single group of Barbary macaques from the Monkey Haven (Isle of Wight, UK; $N = 6$, 21 h of footage) and two groups at the Trentham Monkey Forest (Staffordshire, UK; approximately $N = 140$ of which 34 individuals have been filmed, 10 h of footage). The individuals from the Trentham Monkey Forest were identified by the director of the centre, Sue Wiper, directly from our footage. Those videos have been taken as part of behavioural observations based on focal-animal sampling (*Altmann, 1974*). However, it was not always possible to differentiate the individuals at the Trentham Monkey Forest. The videos comprised a wide range of social and non-social behaviours, taken during natural interactions (Trentham Monkey Forest) or when the animals were performing cognitive tasks (Monkey Haven), which allow us to document a wider range of spontaneous facial expressions. Video footage was cut into short clips of approximately 6 s long; each clip contained either a unique movement or a combination of facial movements. 190 short video clips were created from the original videos. This database is comparable to that used for the development of the MaqFACS (*Parr et al., 2010*).

### Procedure

One trained FACS coder (EJD) certified in human FACS and MaqFACS first coded the 190 video clips using MaqFACS. Spontaneous occurrences of facial movements were identified and coded as events using the Actions Units defined in the MaqFACS manual. Reliability analysis was conducted with another certified MaqFACS coder (CG) through three coding sessions on 60 short video clips out of the 190 clips created (~30% of the footage, containing 18 AU/ADs in total and 235 facial movements) by calculating the ratio of agreement between two coders. A coding session was defined as a round of coding
**Table 1  Agreement per AU between coders.**

| | General agreement | Agreement per AU | | | | | | | | |
| | | AU1+2 | AU41 | AU6 | AU8 | AU9+10 | AU10 | AU12 | AU16 | AU17 |
|---|---|---|---|---|---|---|---|---|---|---|
| *Session1* | 0.73 | 0.64 | 0.50 | 0.67 | 0.00 | 0.80 | 0.96 | 0.77 | 0.94 | 0.00 |
| *Session2* | 0.81 | 0.86 | 0.82 | 0.75 | 0.44 | 1.00 | 1.00 | 0.67 | 0.82 | 0.00 |
| *Session3* | 0.88 | 0.73 | 0.67 | 0.92 | 0.00 | 1.00 | 0.94 | 0.67 | 1.00 | 1.00 |

| | Agreement per AU | | | | | | | | |
| | AU18i | AU18ii | AU25 | AU26 | AU27 | EAU1 | EAU2 | EAU3 | AD181 |
|---|---|---|---|---|---|---|---|---|---|
| *Session1* | 0.53 | 0.40 | 0.96 | 0.77 | 0.75 | 0.67 | 0.00 | 0.67 | 0.67 |
| *Session1.2* | 0.67 | 0.40 | 0.96 | 0.84 | 0.80 | 0.60 | 0.00 | 0.00 | 0.67 |
| *Session2* | 0.67 | 1.00 | 1.00 | 0.89 | 0.94 | 0.89 | 1.00 | 0.67 | 0.67 |

**Notes.**

The agreement score was calculated using a Wexler's score (see formula in text; *Ekman, Friesen & Hager, 2002a*).

on a given number of video clips, resulting in an agreement score between the coders. The coders went through three rounds of coding (i.e., three coding sessions) in order to validate the use of MaqFACS in Barbary macaques. The agreement calculation (Wexler's agreement) was taken from *Ekman, Friesen & Hager (2002a)*, and is the same as used in the previous FACS (human, chimpanzees, gibbons and macaques):

$$\frac{2\,(Number\ of\ AUs\ on\ which\ coder\ 1\ and\ 2\ agreed)}{Total\ number\ of\ AUs\ scored\ by\ the\ two\ coders}$$

After a first round of coding on 40 video clips, the general agreement score was 0.73, which is considered as good agreement in FACS methodology (*Ekman, Friesen & Hager, 2002a*). However, we wanted to systematize the use of MaqFACs to the Barbary macaques, and thus reach an agreement above 0.80. The two coders discussed common coding issues and re-scored the 40 clips. The agreement in this second session was 0.81. In order to generalise the agreement, the coders scored 20 new clips. Agreement between the coders was 0.88 after this third generalisation session with new clips. The detail of the agreement per AU is presented in Table 1. Low agreement scores for some movements are due to the fact that some AUs were extremely rare and/or hard to spot.

Collection of the video material was approved by the University of Portsmouth Ethics Committee and was in compliance with the ASAB/ABS guidelines for the use of animals in Research.

## RESULTS AND DISCUSSION

What follows is a short summary of the facial appearance changes associated with each AU identified in the rhesus macaques, with comparison to the humans, with any differences in Barbary macaques noted (summarized in Table 2).

**Table 2  Summary of AUs in macaques and the difference between species, in comparison to humans.**

| AU | Name | Muscle | Human FACS | MaqFACS | Differences between macaques species |
|---|---|---|---|---|---|
| AU1 | Inner brow raiser | Medial frontalis | ✓ | x | |
| AU2 | Outer brow raiser | Lateral frontalis | ✓ | x | |
| AU1+2 | Brow raiser | Frontalis | ✓ | ✓ | Existence of unilateral AU1+2 in Barbary macaques |
| AU4 | Brow lowerer | CS, DS, Proc | ✓ | x | |
| AU41 | Glabella lowerer | Procerus | ✓ | ✓ | |
| AU5 | Upper lid raiser | Orbicularis oculi | ✓ | x | |
| AU6 | Cheek raiser | Orb. oculi (orbital) | ✓ | ✓ | |
| AU7 | Lid tightener | Orb. oculi (palpebral) | ✓ | ✓ | |
| AU8 | Lips toward each other | Orbicularis oris | ✓ | ✓ | |
| AU9 | Nose wrinkler | Llsan | ✓ | ✓ | |
| AU10 | Upper lip raiser | Levator labii sup | ✓ | ✓ | |
| AU11 | Nasolabial furrow deepener | Zygomaticus minor | ✓ | x | |
| AU12 | Lip corner puller | Zygomaticus major | ✓ | ✓ | |
| AU13 | Cheek puffer | Caninus[a] | ✓ | x | |
| AU14 | Dimpler | Buccinator | ✓ | x | |
| AU15 | Lip corner depressor | Depressor anguli oris | ✓ | x | |
| AU16 | Lower lip depressor | Depressor labii inf | ✓ | ✓ | |
| AU17 | Chin raiser | Mentalis | ✓ | ✓ | |
| AU18[b] | Lip pucker | Orbilaris oris | ✓ | ✓ | AU18i: existence of an open mouth form in Barbary macaques |
| AU20 | Lip stretcher | Risorius | ✓ | x | |
| AU21 | Neck tightener | Platysma | ✓ | x | |
| AU22 | Lip funneler | Orbicularis oris | ✓ | x | |
| AU25 | Lips parted | Various | ✓ | ✓ | |
| AU26 | Jaw drop | Various | ✓ | ✓ | |
| AU27 | Mouth stretch | Various | ✓ | ✓ | |
| AU28 | Lip suck | Orbilaris oris | ✓ | x | |
| EAU1 | Ears forward | Ant. auricularis | x | ✓ | Ear movements are harder to see in Barbary macaques due to their abondant fur |
| EAU2 | Ear elevator | Sup. auricularis | x | ✓ | |
| EAU3 | Ear flattener | Post. auricularis | x | ✓ | |
| AD181 | Lip smacking | | x | ✓ | *Rhesus macaques*: associated with AU18i; *Barbary macaques*: associated with a bared-teeth display |

**Notes.**

CS, corrugator supercilii; DS, depressor supercilii.

Proc procerus, Llsan levator labi superioris alaeque nasi.

[a] The caninus is also referred as the levator aguli oris in humans.

[b] In MaqFACS, the AU18 has been divided into two separate AU codes, AU18i-true pucker and AU18ii-outer pucker (see text).

**Table 3** Exhibited AUs in number of time used and frequencies within the 190 scored video clips.

| Action unit | AU1+2 | AU41 | AU6 | AU8 | AU9+10 | AU10 | AU12 | AU16 | AU17 |
|---|---|---|---|---|---|---|---|---|---|
| Total number | 136 | 56 | 32 | 21 | 7 | 51 | 19 | 50 | 4 |
| Frequency | 20.30 | 8.36 | 4.78 | 3.13 | 1.04 | 7.61 | 2.84 | 7.46 | 0.60 |

| Action unit | AU18i | AU18ii | AU25 | AU26 | AU27 | EAU1 | EAU2 | EAU3 | AD181 |
|---|---|---|---|---|---|---|---|---|---|
| Total number | 26 | 2 | 96 | 57 | 42 | 37 | 5 | 11 | 18 |
| Frequency | 3.88 | 0.30 | 14.33 | 8.51 | 6.27 | 5.52 | 0.75 | 1.64 | 2.69 |

## Movements of the upper face

### AU1+2 (inner and outer brow raiser)

FACS describes independent movements of the inner and outer portion of the brow; however, clear independent movements were not observed in either species of macaques. Thus, the action units AU1 and AU2 have been jointed into a single combined movement AU1+2 describing the raising and lifting of the browline. This movement reveals greater surface area in the underbrow region, so visibility of the underbrow is a particularly salient appearance change for identifying this movement in Barbary macaques (Video S1). Also, depending on the curvature of the brow in the monkey's neutral state, AU1+2 can function to curve the brow into a smooth arc. A final appearance change of AU1+2 is that it can create a bulging of the hair superior to the brow region. This is achieved by the contraction of the frontalis muscle.

In the Barbary macaques, we observed the use of unilateral AU1+2 in three different individuals (Fig. 2) that has not been reported in rhesus macaques. The unilateral AU1+2 was observed in natural context with non-social behaviours when the individuals glanced to the left or right. This movement was not associated with a feeding context only, where a subordinate would have to pay attention to his/her surrounding and look for the presence of dominant individuals. The unilateral AU1+2 was also exhibited when an individual was sitting alone.

This facial movement of the brows was the most common expression observed through the 30 h of footage collected (AU1+2 = 20.30% of the AUs produced within the 190 scored video clips; see Table 3). The frequency of a given AU was calculated as the ratio between the number of times a specific AU was produced within the 190 video clips coded and the total number of AUs produced within the 190 clips. As the sample was *ad libitum*, we deemed this frequency measure more appropriate than signals per hour. Moreover, during direct observations at the Monkey Haven, the individuals presented this behaviour not only when presented with cognitive tasks but also as natural, spontaneously occurring expression.

### AU41 (glabella lowerer)

One of the most conspicuous movements in the FACS is AU4, the brow lowerer (used primarily in human frowning/anger (*Ekman & Friesen, 1978*; *Friesen & Ekman, 1983*)).

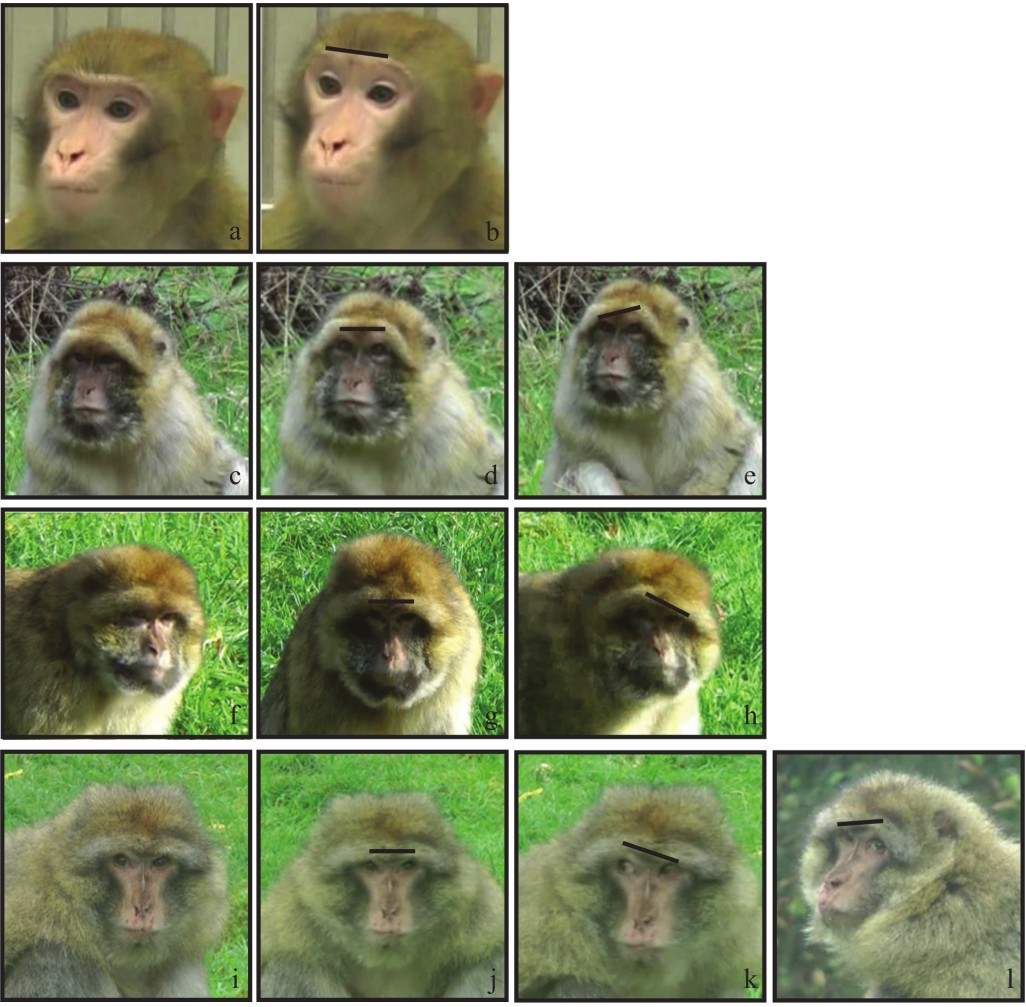

**Figure 2 An illustration of the different AU1+2 in Barbary macaques compared to rhesus macaques.** Rhesus macaques: (A) neutral face; (B) bilateral AU1+2. Barbary macaques: (C), (F) and (I) present neutral faces in three individuals; (D), (G), and (J) present bilateral AU1+2 in the same individuals; finally, (E), (H), (K) and (L) are examples of unilateral AU1+2. The black line show the inclination of the browline: it is straight and up in bilateral AU1+2 and tilts in unilateral AU1+2.

This is achieved by the contraction of three muscles, the corrugator supercilii, depressor supercilii and procerus, which results in both a lowering and medial contraction, e.g., knitting of the brow. FACS also reports individual AUs for the contraction of each muscle independently, AU41 (procerus), AU42 (depressor supercilii) and AU44 (corrugator), although it is very rare that these can be differentiated in humans. Although clear brow lowering movements were observed in rhesus macaques (*Parr et al., 2010*), the medial contraction, or knitting, characteristic of AU4 in humans was not observed. In MaqFACS, *Parr et al. (2010)* reported that the brow lowering movement appears to consist mostly of a medial bulging in the glabella region due to the action of the procerus. Because of this, brow lowering in MaqFACS is specifically identified as AU41 (glabella lowerer).

This movement pulls the brow downward, reducing the visibility of the underbrow and changes the curvature of the brow such that it becomes lowered at the midpoint (Video S3).

In Barbary macaques, the medial bulging in the glabella region is less pronounced or less visible due to the fur of the individuals. However, the lowering of the brow line and the reduced visibility of the underbrow are present. The change in the curvature of the brow is not conspicuous but we do observe clearly the lowering of the central part of the browline.

## Movements of the lower face

### AU9 (nose wrinkle), AU10 (upper lip raiser)

In FACS, the movements of AU9 and AU10 can be tightly coupled. Extreme nose wrinkling, for example, functions to raise the upper lip slightly, and many expressions in humans combine AU9+AU10, as in disgust (*Friesen & Ekman, 1983*). In the rhesus macaques, the AU9, by itself, can be difficult to detect. However, a combined AU9+AU10 has been observed, in addition to the independent action of AU10. Therefore, in describing these two movements, MaqFACS have attempted to describe the appearance changes of each, although in most cases AU9 would be reported in combination with an AU10. The AU9+10 is achieved by the contraction of two muscles, the levatorlabii superioris alaeque nasii and the levatorlabii superioris; the contraction of the latter result in AU10 alone.

In addition to the action of AU10, which pulls the upper lip upward in a smooth arc causing wrinkles and furrows in the infra-orbital triangle, AU9 in MaqFACS functions to pull the nose upward, causing oblique nose wrinkles to deepen. AU9 alone pulls the lateral aspect of the nostril wings upward and medially towards the root of the nose, which causes the nasal groove to deepen.

In Barbary macaques, the conjuncture of AU9 and AU10 is marked by the deepened wrinkles on the infra-orbital furrow and a shortened nasal groove (Video S4). Unique occurrence of AU9 was not observed, as in the rhesus macaques.

### AU12 (lip corner puller)

The function of AU12 is to pull the lip corners back and slightly upward in a movement that produces the homologous expressions of human smiling and the rhesus macaque bared-teeth displays (*Parr et al., 2010*). This is achieved by contracting the zygomatic major. AU12 functions to retract the lips laterally and upwards towards the ears. It narrows and slightly bulges the upper lip, reducing the visibility of the vertical lip ridge and deepening the furrows at the mouth corners (Video S5). In both rhesus and Barbary macaques, AU12 also creates oblique wrinkles and deepens the furrows of the infraorbital triangle, one of its most prominent appearance changes.

### AU16 (lower lip depressor)

The appearance changes associated with AU16 are common to humans and rhesus macaques, despite considerable differences in the morphology of the lips, e.g., thick red reverted lips in humans, and thin lips (appearing to invert) in rhesus and Barbary macaques. Appearance changes associated with AU16 (contraction of the depressor labii inferioris) include lowering the bottom lip to expose the teeth and lower gum. In both rhesus macaques and Barbary macaques, AU16 also causes a slight eversion of the lower

lip, which may cause the inner portion of the lip to appear to thicken slightly. AU16 can also increase the curvature of the lower lip by pulling the medial aspect downwards towards the chin, in contrast to the resting shape of the mouth, in which the medial portion of the lower lip can appear to turn upward.

### AU18i/AU18ii (true pucker and outer pucker)

In FACS, there are two main movements responsible for protruding the lips: AU18—the lip pucker, e.g., when kissing, and AU22—the lip funneler, which pushes the lips outward as if saying the word "flew." In the rhesus macaque, however, lip puckering was attributed to two movements.

The first is a pucker similar to AU18 in humans and referred as AU18i, the true pucker, in MaqFACS. To achieve this movement, three muscles appear to be needed: the orbicularis oris, the incisivii labii superioris, and the incisivii labii inferioris. The AU18i purses the lips medially forward towards each other, narrowing the mouth corners medially, protruding the lips and reducing the mouth aperture in both the horizontal and vertical directions. This movement causes distinct oblique wrinkles to appear extending from the cheek along the length of the upper lip. Because rhesus and Barbary macaques do not have reverted lips, this movement causes the medial portion of the lip to take a scalloped appearance on either side of the midline as the lips are pursed forward. AU18i causes the philtral region to deepen and produces a depression in the medial portion of the lower lip causing it to appear slightly curved (Video S6).

The second lip protrusion movement observed in macaque species contained distinct appearance changes from AU18i, but was also insufficient to be labelled AU22. This movement is instead described as AU18ii, the outer pucker, in which the lips and the lower jaw are pushed forward so as to protrude slightly, causing oblique wrinkles to extend from the cheek along the upper lip. What distinguishes this movement from AU18i is that the furrow between the nose and upper lip (philtral region) is reduced and the lips cinch together at a point distal to the midline causing them to part and appear inflated. Also the movement of the lower jaw helps coding AU18ii. We speculate that the AU18i is produced by the joint action of orbicularis oris, incisivii labii superioris, and incisivii labii inferioris while the movement of AU18ii is produced by contraction of the incisivii portions specifically (incisivii labii superioris and inferioris). This variant was observed as it is described in the MaqFACS in Barbary macaques but was really rare: only two individuals, one in each group, were observed doing an AU18ii (Video S7).

In addition to these, MaqFACS includes an action descriptor, AD181 (lip smacking, contraction of the orbicularis oris), to denote tightening of the lips together followed by a rapid opening and parting motion, which is a common facial movement associated with AU18i and the lip-smacking expression of the rhesus macaque. In Barbary macaques, the 'lip smacking' is present also associated with a bared-teeth display. Hence, the Barbary macaques present both a lip-smacking and a teeth chattering (Wiper & Semple, 2007), that was not reported in rhesus macaques.

## Ear movements
### EAU1 (ears forward), EAU2 (ears elevator), EAU3 (ears flattener)

Unlike many mammals, humans lost the ability to move their ears independently. Therefore, FACS does not contain descriptions of ear movements. Among rhesus and Barbary macaques, three prominent and independent ear movements are described. EAU1 (ears forward) functions to push the ears forward towards the face via the contraction of the anterior auricularis, increasing the visibility of the ear if viewed from a frontal orientation, but reducing the visibility of the ear if viewed in profile (Video S8). EAU2 (ear elevator) pulls the ears superiorly towards the top of the head; this movement if still present in the Barbary macaques appeared less common than in the rhesus macaques or less easy to observe due to the facial fur of the Barbary macaques (Video S2). This results from the contraction of the superior auricularis. Finally, EAU3 (ear flattener) pulls the ears towards the back of the head, flattening them against the skull, by contracting the posterior auricularis. This may reduce the visibility of the ears if viewed from a frontal orientation, but increase the visibility of the ears if viewed in profile (Video S3).

It should be noted that coding specific EAUs can be very difficult if the neutral position of the ears is unknown, e.g., an EAU1 may actually be the release of an EAU3. Moreover, fighting among macaques often injures the ears, and these injuries can reduce the visibility of the pinnae that are required to denote the appearance changes described above. Also, the Barbary macaques present an abundant facial fur, which makes it even more difficult to code subtle ear movements (Fig. 1). Thus, in the MaqFACS, it is recommended that users code an EAD, Ear Action Descriptor, to denote movement of the ears without specifying its muscular basis unless there is clear sufficient evidence about the neutral position of the ears to justify a specific EAU code.

## General discussion

Overall, the agreement score between the two coders was 0.88, which is excellent agreement in FACS methodology (*Ekman, Friesen & Hager, 2002a*). Thus, MaqFACS can be used to code Barbary macaque facial movements reliably. Similarity in facial movement is likely due to the great degree of conservation in basic facial muscles among macaques (*Burrows, 2008*).

Barbary macaques exhibit a range of facial movement highly similar to rhesus macaques and close to that of chimpanzees and humans. Adapting the FACS systems revealed remarkable similarities in the facial movement across related species, with however some species-specific movements. The Barbary macaques tend to display an important amount of AU1+2 (over 20% of the total number of AUs scored in the video clips; Table 3) and are able to present unilateral brow movements (unilateral AU1+2; Fig. 2). Moreover, Barbary macaques seem to use AU18i more often than AU18ii, even in threatening situation. On the contrary, rhesus macaques seem to be using more AU18ii to threaten than AU18i. Regarding the ear movements, EAU2 seem rare with a predominance of EAU1, which could play an important part in watching one's surrounding and being aware of who is around. Those ear movements are a prominent feature in macaques, including lipsmack

and bared-teeth displays, and therefore are presumed to play an important role in social communication (*Partan, 2002*; *Van Hooff, 1962*; *Van Hooff, 1967*).

The current study presents some limitations. Neither facial muscles dissection nor intramuscular electric stimulation studies were conducted for ethical and practical reasons. Therefore, it is possible that missing steps caused us to draw partial conclusions about facial movements. However, the Barbary macaques are the most ancient taxon and strongly resemble the rhesus macaques in appearance (*Fa, 1989*; *Fooden, 2007*). It is thus likely that the conclusions drawn for rhesus macaques apply for Barbary macaques. The rareness of some Action Units (AUs) can explain some of the low agreement scores (Table 2). For instance, AU8 if present is not a common movement unlike AU10; it is thus harder to obtain a high level of agreement on such rare, discreet, events. The biggest difference between those two species seems to be the use made of facial movements, not the facial movements themselves. Future research should broaden this study and explore differences in facial expressions in different species of macaques and examine the functional significance of reported differences according to their social style and ecology.

## ACKNOWLEDGEMENTS

The authors thank the staff of the Monkey Haven for granting us access to the macaques; and Sue Wiper and the guides from the Trentham Monkey Forest for their warm welcome and permission to collect footages of their animals.

### Funding

This project was funded by a University of Portsmouth Department of Psychology Small Grant to BMW, and a British Academy/Leverhulme Trust Small Research Grant to JM, BMW and AMB. MJ is funded by an EU Marie Curie fellowship (FP 7, project 'Macacognitum' no 623908). The funders had no role in study design, data collection and analysis, decision to publish, or preparation of the manuscript.

### Grant Disclosures

The following grant information was disclosed by the authors:
University of Portsmouth Department of Psychology Small Grant.
British Academy/Leverhulme Trust Small Research Grant.
EU Marie Curie fellowship: 623908.

### Competing Interests

The authors declare there are no competing interests.

### Author Contributions

- Églantine Julle-Danière conceived and designed the experiments, performed the experiments, analyzed the data, contributed reagents/materials/analysis tools, wrote the paper, prepared figures and/or tables, reviewed drafts of the paper.

- Jérôme Micheletta conceived and designed the experiments, wrote the paper, prepared figures and/or tables, reviewed drafts of the paper.
- Jamie Whitehouse and Marine Joly contributed reagents/materials/analysis tools, reviewed drafts of the paper.
- Carolin Gass performed the experiments, analyzed the data, contributed reagents/materials/analysis tools, reviewed drafts of the paper.
- Anne M. Burrows reviewed drafts of the paper, provided anatomical expertise.
- Bridget M. Waller conceived and designed the experiments, wrote the paper, reviewed drafts of the paper.

### Animal Ethics

The following information was supplied relating to ethical approvals (i.e., approving body and any reference numbers):

Collection of the video material was approved by the University of Portsmouth Ethics Committee and was in compliance with the ASAB/ABS guidelines for the use of animals in Research. Research permission was given by the Monkey Haven (Newport, Isle of Wight, UK) and Trentham Monkey Forest (Stoke-on-Trent, Staffordshire, UK).

### Supplemental Information

Supplemental information for this article can be found online at http://dx.doi.org/10.7717/peerj.1248#supplemental-information.

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
