# Peer review of "MaqFACS (Macaque Facial Action Coding System) can be used to document facial movements in Barbary macaques (Macaca sylvanus)"

_PeerJ, doi:10.7717/peerj.1248_

## Round 0.1 · original submission · Major Revisions

You will see that the reviewers' opinions of the MS varied to a great degree. I am willing to invite a revision of the paper provided that you can satisfy the reviewers' concerns. Even the more positive reviewers had substantial concerns that need to be addressed before the MS can be considered publishable in PeerJ. For instance, you need to clarify the issues of individual identification. I will not reiterate each of the reviewers' concerns as they are clearly stated. Reviewer 3 is concerned that steps were missed in the validation process. As I read it, the first two steps were used to create the FACS system initially and may not necessarily need to be repeated in order to generalize the system to new species. Has this been done when developing a system for hylobates, orangutans, cats and dogs? However, this needs to be clarified and better justified in the revision.

I also appreciate the concern that the results do not provide answers to testable hypotheses. However, as I understand it, the aim of the study is to validate whether the FACS system for macaques can be used in Barbary macaques. Perhaps the affirmative response to this needs to be more clearly stated. Currently the results read as a list of descriptives. Perhaps more about why these descriptions are important - what do they tell us - would help. You could focus more explicitly on differences in expression and use between species in each section.
I have a few comments of my own. Why is the reliability measure 2X the number of rater agreements divided by the total number of agreement (line 145)? This seems an extremely liberal measure of agreement.
If the use of facial movements are quite different across species, how can we be sure that the same movements represent the same underlying condition? Please expand on the limitations.

·

Basic reporting

The paper is generally well written, clearly structured and arguments wel supported with appropriate references. Just a few grammatical / typographical errors:
Line 40 – what does ‘their’ refer to?
Line 87. Full stop missing
Line 131 change ‘where’ to ‘were’
Line 344 change researches to research
Line 463 change ‘all’ to ‘whole’
Table 3 – what is the unit for frequency (signals/hour?)

Experimental design

The study seems well conducted - I would just like a little more information in a few places:

Line 128 – If you can’t identify individuals, it seems strange to find in the results that certain AUs were seen in 3 individuals. It would be good to know how widespread the observed AU use is across individuals (females/males) etc – can you give some information on where the 190 analysed clips come from so the degree of uncertainty on individual identity is known (e.g. 30 known individuals (15 males, 15 females) contribute between 1 and 5 clips and 10 unknown individuals contribute a total of 25 clips)

Line 141 – please explain what a ‘coding session’ and a ‘clip’ constitutes – the explanation of the stepped IOR is confusing without this.

Line 142- please state min and max values for the Wexler agreement. I assumed the max value was 1, but there is a value of 1.13 in table 1.

Validity of the findings

No comments

Reviewer 2 ·

Basic reporting

This study aims to show that the Facial Action Coding System (FACS) developed for rhesus macaques can be used in another species, the Barbary macaques. This study appears satisfactory to me. Since it does not include statistical results, my review will be unusually short. I have only a few requests for clarity. Note however that I am not an expert in the Facial Action Coding System.

- l. 113: do not go to line, this sentence should belong to the previous paragraph.

- l. 175, 199, 237, 249, 260, 297: rhesus macaques? please specify.

- l. 182: this formulation is unclear.

- l. 243: without statistics, it is difficult to understand what means "no major differences".

- l. 297: again, please specify which macaques.

Experimental design

No problem

Validity of the findings

This study represents a significant methodological advance in the study of primate facial expressions.

Reviewer 3 ·

Basic reporting

Acceptable - no problems

Experimental design

This paper does not have an experimental hypothesis; the purpose of the study is stated as "we wanted to test whether [...] MaqFACS developed for rhesus macaques could be used to code facial movements in Barbary macaques" (l. 133-115). Speculations are made that differently despotic social systems may result in differentially expressive facial gestures, yet no clear predictions are made, and the results do not discuss whether the findings confirm or disprove any potential hypotheses. So, I am not sure what to make of this. The authors could have equally asked whether the orangutan FACS system could be used to code Barbary macaque facial expressions (on the basis of a distant common ancestor); some similarities and differences would have been noted, and we'd have a roughly similar paper with a bit of a non-conclusion. So, while perhaps technically sound (see comments below), this papers falls short on producing a meaningful finding.

Validity of the findings

The authors describe a rigorous coding system with high degrees of reliability (but note: I am unclear what "after three coding session", l. 152, refers to). What is more concerning is that the authors outline that the FACS system development relies on a 3 step procedure: dissection to identify muscles, intramuscular stimulation to confirm surface movements of facial muscles, and confirmation of muscles contractions from spontaneous behavior (l. 73-79). For this particular study, however, steps 1 and 2 were skipped and the authors only report data from step 3, spontaneous behavior. Thus, any conclusions is in part speculation and these findings cannot be regarded as robust until the first two steps have been undertaken. I fully understand that collecting these data can be difficult, and it is quite likely that Barbary macaque facial musculature is very similar to rhesus facial musculature, but I believe for the science to be robust, we cannot take shortcuts like this.

---

## Round 0.2 · Minor Revisions

Thank you for clarifying some of the issues raised by the reviewers and by me in the previous round of comments. I am prepared to accept your MS but first, I will need you to attend to the following grammatical issues. When you have corrected these issues please upload a correct copy of the MS so that I can render an official decision of accept. Thank you.

Please insert a space between “species” and “phylogenetically” on line 13, and between “genus” and “(Purvis” on line 115.

I am confused by the statement, “This movement was not associated with a feeding context only” on line 203. Please clarify.

Also please clarify the sentence on lines 206-208. I am confused by “the number to times a specific AU was produced”.

On line 209 should “tis” be “this”?

There are some odd square symbols following sub-headings. Please omit if unintended.

On line 333, “result” should be “results”.

I don’t think that the inclusion of “rhesus or Barbary” is necessary on line 340 as this statement is probably true of all macaques.

On line 368, should “conclusion” be pluralized?

On line 372, should “agreements” not be pluralized?

---

## Round 0.3 · accepted · Accept

Thank you for taking care of these additional very minor details. I am happy to now accept your paper for publication in PeerJ.